# Exploiting Differences in Heme Biosynthesis between Bacterial Species to Screen for Novel Antimicrobials

**DOI:** 10.3390/biom13101485

**Published:** 2023-10-06

**Authors:** Laurie K. Jackson, Tammy A. Dailey, Brenden Anderle, Martin J. Warren, Hector A. Bergonia, Harry A. Dailey, John D. Phillips

**Affiliations:** 1Department of Internal Medicine, Division of Hematology, University of Utah, Salt Lake City, UT 84112, USA; lj.datatransfer@gmail.com (L.K.J.); hector.bergonia@hsc.utah.edu (H.A.B.); 2Department of Microbiology, Department of Biochemistry and Molecular Biology, University of Georgia, Athens, GA 30602, USAhdailey@uga.edu (H.A.D.); 3WhiteTree Medical, 10437 S Jordan Gateway, South Jordan, UT 84095, USA; brenden@whitetreemedical.com; 4Quadram Institute Bioscience, Norwich Research Park, Norwich NR4 7UQ, UK; martin.warren@quadram.ac.uk

**Keywords:** antimicrobial, heme biosynthesis, screen, monoderm, diderm

## Abstract

The final three steps of heme biogenesis exhibit notable differences between di- and mono-derm bacteria. The former employs the protoporphyrin-dependent (PPD) pathway, while the latter utilizes the more recently uncovered coproporphyrin-dependent (CPD) pathway. In order to devise a rapid screen for potential inhibitors that differentiate the two pathways, the genes associated with the protoporphyrin pathway in an *Escherichia coli* YFP strain were replaced with those for the CPD pathway from *Staphylococcus aureus* (SA) through a sliding modular gene replacement recombineering strategy to generate the *E. coli* strain *Sa*-CPD-YFP. Potential inhibitors that differentially target the pathways were identified by screening compound libraries against the YFP-producing *Sa*-CPD-YFP strain in comparison to a CFP-producing *E. coli* strain. Using a mixed strain assay, inhibitors targeting either the CPD or PPD heme pathways were identified through a decrease in one fluorescent signal but not the other. An initial screen identified both azole and prodigiosin-derived compounds that were shown to specifically target the CPD pathway and which led to the accumulation of coproheme, indicating that the main target of inhibition would appear to be the coproheme decarboxylase (ChdC) enzyme. In silico modeling highlighted that these inhibitors are able to bind within the active site of ChdC.

## 1. Introduction

Many biological processes depend upon heme, an iron-containing porphyrin (iron protoporphyrin IX), as a cofactor, in part because of the ability of the iron to be in the 2+ or the 3+ state. Heme is an essential prosthetic group for the function of diverse proteins including cytochromes, hemoglobins, peroxidases, catalases, and sensors that bind diatomic gases [1,2]. Heme also affects multiple aspects of bacterial pathogenesis, including the ability of *Mycobacterium* and *Campylobacter* to scavenge reactive nitrogen species produced by the host immune system [3,4] and, in *Staphylococcus*, the ability to modulate virulence [5,6]. While the ability to synthesize heme is not ubiquitous in nature, there are few organisms that do not synthesize heme and even fewer that lack heme altogether. Inhibition of heme biosynthesis pathway enzymes by either chemical (e.g., succinyl acetone or N-methylmesoporphyrin) or genetic means results in either cell death or greatly diminished growth. For example, the deletion of ferrochelatase (*cpfC*) in *M. tuberculosis* is lethal [7] and the deletion of porphobilinogen synthase (*pbgS*, aka “*hem B*”) in *S. aureus* results in small colony variants that do not establish infections in vivo [8,9]. These and other findings demonstrate that heme synthesis is essential for the pathogenicity of many bacteria and illuminate the validity of targeting this pathway for new antimicrobial compounds [10,11]. As a proof of concept for targeting heme synthesis to inhibit organismal growth, the classical heme biosynthetic pathway in plants has been targeted and several herbicides which inhibit plant protoporphyrinogen oxidase are now widely used [12].

For over half a century, the accepted belief was that the heme biosynthetic pathway was conserved among all organisms with all metabolic intermediates invariable. Thus, target compounds toxic to a prokaryotic infectious agent would be equally toxic to a metazoan host. This so-called “classic pathway” possesses alternative enzymes at two steps to accommodate aerobic vs. anaerobic lifestyles, i.e., the decarboxylation of coproporphyrinogen III to protoporphyrinogen IX (aerobic: CgdC, anaerobic: CgdH) [13,14,15] and the oxidation of protoporphyrinogen IX to protoporphyrin IX (aerobic: PghdH1, mixed: PgdH1/2) [15,16,17] (Figure 1). However, the only chemical variation known to occur was for the synthesis of the first committed intermediate, 5-aminolevulinic acid (ALA) [15,18]. The “C_4_”, or Shemin, path employs a single enzyme 5-aminolevulinic acid synthase (AlaS) that utilizes glycine and succinyl-CoA as substrates, and this pathway is found in metazoans and a few bacteria. The “C_5_” path relies upon two enzymes, glutamyl tRNA reductase (GtrR) and glutamate-1-semialdehyde-2,1-aminomutase (GsaM), to convert glutamyl-tRNA to ALA [15,19]. This pathway is found in plants and among prokaryotes, the C_5_ pathway predominates with few microorganisms possessing the C_4_ path.

The first experimental hint that the “classical” protoheme synthetic pathway was not universal came in 1998 with the postulation of a “primitive” pathway in *Desulfovibrio vulgaris* [20]. This pathway, later named the “alternative heme biosynthesis” path (or ahb), is found only in sulfate-reducing and heme-synthesizing archaea [21,22,23]. In the ahb pathway, siroheme, synthesized from the pathway intermediate uroporphyrinogen III, is metabolized by demethylation, deacetylation, and then decarboxylation to yield protoheme. In 2015, it was demonstrated that neither the classical or ahb pathways exist in bacteria from the Firmicutes (monoderm bacteria, e.g., *Staphylococus*) and Actinobacteria (e.g., Mycobacteria, e.g., *M. tuberculosis*) phyla [24] and that these organisms are incapable of synthesizing protoporphyrin IX or utilizing it as a pathway intermediate. These organisms have an altered set of the three terminal reactions where coproporphyrinogen oxidase (CgoX) converts coproporphyrinogen III to coproporphyrin III, and coproporphyrin ferrochelatase (CpfC) catalyzes the insertion of ferrous iron into coproporphyrin (rather than protoporphyrin), resulting in the formation of coproheme III (Figure 1). A newly identified enzyme, coproheme decarboxylase (ChdC) [25,26], then decarboxylates coproheme III to generate protoheme IX. This pathway is named the coproporphyrin-dependent (CPD) heme synthetic pathway. These findings created a paradigm shift for the field of heme biosynthesis and revealed a potential antibiotic target, given that ChdC is unique to Firmacutes and Actinobacteria and essential for heme synthesis by these organisms.

Herein, we describe an in vivo screening platform designed to identify compounds that specifically inhibit the growth of CPD-dependent organisms. This platform provides an efficient and robust primary screen that contains appropriate internal controls. Initial experiments that employed this screening platform identified a number of compounds that are putative inhibitors of the CPD pathway-specific enzymes.

## 2. Materials and Methods

### 2.1. Materials

Matched strains MC4100-YFP and MC4100-CFP, that differ only in whether they have yellow (*yfp*) or cyan (*cfp*) alleles of green fluorescent protein integrated into the *galK* locus of their genomes [27,28], were kindly gifted to us by Roy Kishony of Harvard Medical School and Harvard University School of Engineering and Applied Sciences. pKIKOLacZKm [29] was purchased from Addgene.org. pKD78 was obtained from the Coli Genetic Stock Center at Yale [30]. *S. aureus cgoX*, *chdC*, and *cpfC* genes, used for gene replacements, were PCR-amplified from DNA extracted from MRSA 177 (ATCC), as a gift from Dr. Hillary Crandall. Small Molecule Chemical Libraries used in screens were obtained through the University of Utah’s Drug Discovery Core Facility. Prodigiosin (NSC# 47147) and butylcycloheptylprodiginine hydrochloride (NSC# 247562) were originally identified by screening the Natural Products Set IV Library (a subset of NCI/NIH’s DTP Open Repository collection). Additional material for NSC# 47147 and 247562 was obtained from the NCI Open Chemicals Repository, Developmental Therapeutics Program, Division of Cancer Treatment and Diagnosis, National Cancer Institute, National Institutes of Health (NIH). High-Fidelity Platinum Taq (ThermoFisher Scientific, Waltham, MA, USA) and Q5 High-Fidelity DNA Polymerase (New England Biolabs, Ipswitch, MA, USA) were used to amplify *S. aureus* heme genes. Hemin (Sigma) 1000× stock solutions were made at 5 mg/mL in 0.1 M NaOH.

### 2.2. Recombineering of Sa-CPD-YFP

We replaced the *pgdH1*, *cgdC*, and *ppfC* genes in the *E. coli* strain MC4100-YFP with *S. aureus cgoX*, *chdC*, and *cpfC* genes, respectively, to generate a triple replacement strain (diderm → monoderm terminal heme biosynthesis genes). To replace these genes, leaving no additional modifications behind, we used a method described by Li et al. [31]. This method involved multiple rounds of two-step gene replacements. In step one, an *E. coli* heme gene was replaced with a selection/counter-selection *tetA-sacB* cassette by homologous recombination (Figure 2), and recombinants were identified by tetracycline resistance. In step two, we replaced the *tetA-sacB* cassette with the corresponding monoderm heme gene. The *sacB* gene allows counter-selection to select against bacteria that retain the *sacB* insert. The *sacB* gene product converts sucrose to levan, which is toxic to *E. coli*. The *tetA* gene causes cells to become sensitive to fusaric acid, and fusaric acid was also useful as a counter-selection agent. Because the heme biosynthesis pathway is essential to these bacteria, all recombinant *E. coli* were necessarily grown in the presence of added hemin (5 µg/mL) until all three ultimate diderm heme biosynthesis genes (*pgdH1*, *cgdC*, and *ppfC*) had been replaced with the three ultimate monoderm pathway genes (*cgoX*, *chdC*, and *cpfC*) (Figure 3). The third replacement was performed without a *tetA-sacB* intermediate. Instead, recombinants were selected for the ability to grow on LB agar with no added heme. All clones with gene replacements were re-streaked on fresh plates and fully sequenced (amplified by PCR and the PCR products were Sanger-sequenced) (see Appendix A).

Genes were replaced in the following order: (1) *tetA-sacB* replaced *pgdH1* (in MC4100-YFP *E. coli*), (2) *S. aureus* cgoX replaced *tetA-sacB*, (3) *tetA-sacB* replaced *cgdC*, (4) *S. aureus chdC* replaced *tetA-sacB*, and (5) *S. aureus cpfC* replaced *ppfC*, selected on an LB Amp plate with no added hemin (Figure 4). The *E. coli pgdH1* and *S. aureus chdC* genes both begin with a non-traditional start codon GTG (Figure 4). This codon was kept as GTG. Colonies were streaked out on fresh pates to ensure clonality, and all gene replacements were verified by PCR analysis and by sequencing the introduced genes and surrounding areas to rule out unintended changes that may have occurred during the homologous recombination process. qPCR analysis was also used to confirm the expression, or lack of expression, of the targeted genes.

We further replaced *cgdH* with a kanamycin resistance marker (kmR) (*aphA*) surrounded by flippase recognition target sites (FRTs). frt-kmR-frt was amplified from the plasmid pKIKOLacZKm and used to replace *cgdH* by homologous recombination as above. *Sa*-CPD-YFP, used in our screens, retains the kmR marker. We used MC4100-CFP as the parental control in our screens; however, we subsequently generated a version of MC4100-CFP (“WT-CFP”) that had the *cgdH* gene knocked out (“WT-CFPΔN-A”), which could be useful in future screens.

#### 2.2.1. Preparation of Linear DNA Fragments for Homologous Recombination

High-Fidelity Platinum Taq (ThermoFisher Scientific, Waltham, MA, USA) or Q5 High-Fidelity DNA Polymerase (NEB) was used to PCR-amplify the replacement cassettes. Chimeric primers were designed that were composed of regions complementary to the amplicon (3′ end) and with regions of homology to the desired integration site in the target genome (5′ end) (Figure 3 and see Appendix A). Regions of homology to the target gene were approximately 50 bp in length at each end of the replacement cassettes. When recombination proved to be inefficient, longer (~500 bp) arms were generated for integration cassettes using three-part PCR (Appendix A). For three-part PCR, the basic integration cassette was generated using two primers, as described above. Then, additional primers were used to create longer amplicons homologous to the target site, on both 5′ “A” and 3′ “B” sides of the target area, that overlapped slightly with the regions of target homology on the basic integration cassette “C”. Each PCR was performed separately, purified by agarose gel, cleaned up using a Gel extraction kit (Qiagen, Redwood City, CA, USA), and eluted in water. A fourth PCR reaction used all three cleaned PCR products (A, B, and C) as templates for amplification, with the outermost two primers, and the longer integration cassette was separated by agarose gel and cleaned as above for electroporation into electrocompetent cells.

#### 2.2.2. Making MC4100-YFP (Amp^R^) Efficient for Homologous Recombination

We transformed cells with a plasmid pKD78 (CmR), expressing λ red recombination genes, which make *E. coli* K-12 highly efficient at recombination. The λ red genes are driven by an arabinose-inducible promoter. We used a Zymo Research Mix & Go kit (Zymo Research, Irvine, CA, USA) to generate chemically competent MC4100-YFP cells, which were then transformed with pKD78. Because pKD78 is temperature-sensitive, *E. coli* carrying pKD78 were typically cultured at 30 °C.

#### 2.2.3. Preparation of Electrocompetent Cells and Section Media for Recombination

Cells were generated as described [31] with adaptations. All selection media for heme pathway knockouts contained 5–15 µg/mL hemin. To generate electrocompetent cells, cultures of cells carrying the pKD78 plasmid were grown overnight at 30 °C in LB with 10 µg/mL chloramphenicol (Cm) and 50 µg/mL ampicillin (Amp). Overnight cultures were diluted 1/100 in fresh media containing Cm and cultured at 30 °C for one hour prior to the addition of 20 mM arabinose. When the cells reached an OD 600 of 0.4 to 0.5, culture flasks were placed in a 42 °C water bath with gentle swirling for 15 min, then removed from the bath and swirled in an ice water bath to cool, and left on ice for 10 min. Cultures (50 mL) were centrifuged 1800× *g* at 4 °C and the cell pellet was resuspended in 35 mL of ice-cold water (twice). Cells were pelleted twice more and resuspended in 10 mL and then 0.3 mL of ice-cold 10% glycerol. Cells were used directly for electroporation or flash-frozen for future use (−80 °C). An amount of 2–5 µL of replacement cassette DNA (up to 200 ng) was mixed with 50 µL cell aliquots and electroporated in 0.1 cm cuvettes using a BioRad Gene Pulser Xcell (Bio-Rad Laboratories, Hercules, CA, USA) (1.8 (or 2.5) KV, 25 uF, 200 ohms, ~5 msec). Following DNA electroporation, cells were transferred to LB with 5 µg/mL hemin and allowed to recover for four hours in a shaking incubator at 30 °C, and then plated to the appropriate selection or counter-selection media.

#### 2.2.4. Selection of *tetA-sacB* Integrants

Cells were plated onto LB agar plates containing 12 µg/mL tetracycline (Tc), 5 µg/mL hemin, and +/− 10 µg/mL chloramphenicol (Cm). Plated cells were incubated at 30 °C to maintain the pKD78 plasmid for subsequent rounds of gene replacement. However, it was often the case that recombinant cells did not grow unless the temperature was raised to 34 or 37 °C. The recipe used for LB agar plates was 10 g of tryptone, 5 g of yeast extract, 5 g of NaCl, and 15 g of agar per liter.

#### 2.2.5. Counter-Selection (Replacement of *tetA-sacB*)

Cells were plated onto agar plates containing: 15 g of agar, 4 g of tryptone, 4 g of yeast extract, 8 g of NaCl, 8 g of NaH_2_PO_4_·H_2_O, and 100 g of sucrose per liter with the following: 0.8 mM ZnCl_2_, 24 µg/mL fusaric acid, 5 µg/mL hemin, and +/− 10 µg/mL Cm. The agar, tryptone, and yeast extract were autoclaved separately in 600 mL of water, NaCl and NaH_2_PO_4_.H_2_O were autoclaved together in 200 mL of water, and 100 g of sucrose was autoclaved in 160 mL of water. After the autoclaved solutions had been cooled to 55 °C and mixed, ZnCl_2_ was added from a filter-sterilized 25 mM stock, fusaric acid from a 48 mg/mL stock (in ethanol), and hemin from a 5 mg/mL stock. The solutions were brought to 1 L with autoclaved water. We used more sucrose (10%) than typically recommended (6%) for this method due to high background, but later reduced sucrose to 6% in multi-gene replacement knockouts. Cells were incubated on the counter-selection plates for two days at 42 °C then moved to 30 °C or 37 °C to incubate while waiting for the appearance of slowly growing colonies.

### 2.3. qPCR

1 µg of RNA, extracted from cell pellets derived from log phase bacterial cultures, was used to generate cDNA using Thermo Scientific Maxima H Minus Reverse Transcriptase kit components (ThermoFisher Scientific, Waltham, MA, USA). SYBR Green qPCR Master Mix (ThermoFisher Scientific, Waltham, MA, USA) was used in quantitative PCR reactions with primers as shown in Appendix A, using an annealing temperature of 55 °C. All qPCR primers were designed using Primer3Plus.com, with the exception of 16S primers [32].

### 2.4. The Screen

A mixture of WT-CFP (wild-type MC4100 *E. coli* expressing CFP) and *Sa*-CPD-YFP (matched YFP-expressing *E. coli* strain with *S. aureus* terminal heme genes (CgoX, CpfC, and ChdC) replacing *E. coli* heme genes (PgdH1, CgdC, and PpfC)) was used in a competition viability assay to identify compounds that specifically target CPD heme biosynthesis pathway enzymes, unique to monoderm bacteria. The screen was initiated by the growth of each strain in liquid culture to mid-log phase in minimal media; cells were quantified, pre-mixed at a stoichiometric ratio, added to wells of 96-well plates containing drug or vehicle, and then incubated with shaking at 30 °C for 16 h. The fluorescence of YFP and CFP was measured, using Bio-Tek Synergy4 and Bio-Tek Synergy Neo2 plate readers (Bio-Tek Instruments, Winooski, VT, USA) at excitation/emission (ex/em) = 445/475 nm (CFP for the wild-type PPD pathway) and 500/536 nm (YFP for *Sa*-CPD-YFP) to quantify the growth of wild-type and *Sa*-CPD-YFP bacteria in each well (Figure 5). Wells with vehicle and media only (no bacteria) were used to determine background. Readings from each well/condition had the background subtracted and wells containing test substances were normalized to the respective fluorescence signals in vehicle-treated wells containing bacteria. As a control, we set up matching screen plates that contained 7 µM of hemin in the media (the product of the targeted pathway) to ensure that the inhibition was pathway-specific. We pursued leads only if they both selectively targeted the *Sa*-CPD-YFP strain and not the wild-type bacteria and if the inhibition was also rescued by hemin.

Plates were set up by adding: 100 µL of M9 media, 2 µL of compound in DMSO (or DMSO only), and 100 µL of bacteria (see below) in M9 media +/− 10 µg/mL hemin (2×). Compounds for screens were typically taken from stock plates containing 1 mM compound in DMSO, resulting in a top concentration of 10 µM. M9 media has low background fluorescence and was used to culture bacteria for our screen. Our M9 media contained 6 g/L Na2HPO4, 3 g/L KH2PO4, 1 g/L NH4HCl, 0.5 g/L NaCl, 1 mM MgSO4, 0.1 mM CaCl2, 0.1% glucose, 0.00165% Thiamine, and 0.2% casein amino acids.

To prepare bacteria cultures for the screen, *Sa*-CPD-YFP and wild-type WT-CFP strains were grown separately in M9 media containing 50 µg/mL Amp in a shaking incubator at 30 °C overnight. The next day, these cultures were serially diluted in M9 media (no antibiotic) and incubated at 30 °C to obtain cultures that were in log phase growth by late afternoon (OD 600 nm ~0.2). *Sa*-CPD-YFP bacteria grow more slowly than WT-CFP, so the initial inoculum for the screen was created with a higher starting concentration of *Sa*-CPD-YFP compared to WT-CFP. To prepare a stoichiometric mixture of *Sa*-CPD-YFP and wild-type WT-CFP strains, absorbance readings from the log phase cultures were taken at OD 600 nm and then bacteria were diluted into M9 media to achieve calculated final “concentrations” of OD 600 0.1 for *Sa*-CPD-YFP and 0.02 for WT-CFP. This initial WT-CFP + *Sa*-CPD-YFP mixture was diluted another 1000× in M9 media prior to addition to screen plates. The plates were covered with lids and taped around the edges using masking tape to prevent drying, and incubated in a shaking incubator at 30 °C in the dark for 16 h. A freezer rack made for 96-well microtiter plates was attached to a shaking incubator for this purpose.

### 2.5. Calculations of Z’ Factors for the screen

Antibiotic (such as spectinomycin) was used as an inhibitor to calculate Z′ factors for the screen (Equation (1)). Z′ factor is defined as:(1)$$Z^{’}=1-\left[\frac{3\sigma n+3\sigma p}{\left|µn-µp\right|}\right]$$
where 3σn, 3σp, µn, and µp are the standard deviations (σ) and averages (µ) for the negative (n) and positive (p) controls. For a well-defined signal window, worthy of a screen, Z′ should be greater than 0.5 [33]. For this calculation, a 96-well plate was used, and the growth conditions and dilutions were identical to the screen. Column 1 was media control, columns 2–6 were set up as the negative control (*Sa*-CPD-YFP and WT-CFP mixture, no antibiotic), and columns 7–11 were set up as the positive control (*Sa*-CPD-YFP and WT-CFP mixture with 50 µg/mL spectinomycin). A Z’ factor was calculated for both the YFP (ex/em = 500/536 nm, *Sa*-CPD-YFP) and CFP (445/475, WT-CFP) channels.

### 2.6. Titration of Screen Hits

A serial five-fold dilution of the compound of interest was generated and used to treat mixtures of wild-type and mutant strains either in the absence or presence of hemin, using a 96-well format and assay conditions similar to those used in the identifying pilot screen. Plates were read at YFP and CFP wavelengths and plotted as a percent of DMSO-treated control.

### 2.7. Determination of Background Signal for Prodiginine Family Compounds

To determine whether butylcycloheptylprodiginine, prodigiosin, or other prodiginines could interfere with YFP or CFP fluorescence signals from an *Sa*-CPD-YFP/WT-CFP mixed culture grown under primary screen conditions, serial dilutions of the compound were mixed into dedicated control wells in 96-well plates after the cells had grown for 18 h at 30 °C in the shaking incubator, shortly before taking fluorescent readings.

### 2.8. UPLC Assays for Heme and Porphyrins

A Waters Acquity (Milford, MA, USA) ultra-performance liquid chromatography (UPLC) system equipped with a reverse phase C18 column, a photodiode array detector (PDA), and a fluorescence detector was used to separate and quantify concentrations of total heme (as hemin, the chloride salt of oxidized heme), protoporphyrin IX (PPIX), and intermediate porphyrins. Heme(s) and PPIX were extracted from sonicated cell homogenates containing about 10 mg/mL protein by mixing with four volumes of an extraction solvent (4:1 ethyl acetate:glacial acetic acid) and vortexing for 60 s. The mixture was then centrifuged at 16,000× *g* for one minute, and the resulting upper organic layer containing heme and PPIX was analyzed by UPLC. The absorbance of the non-fluorescent hemin was monitored at 398 nm, while PPIX was monitored using fluorescence (excitation 404 nm, emission 630 nm) [34,35]. Intermediate porphyrinogens from heme biosynthesis were oxidized to fluorescent porphyrins by adding an equal volume of 3 M HCl to sample homogenates, incubated for 30 min at 37 °C, and then centrifuged at 16,000× *g* for 10 min. These clarified samples were injected into the UPLC and visualized with PDA and a fluorescence detector set at 404 nm excitation and 618 nm emission. Peak identities were confirmed via comparison with retention times of heme, intermediate porphyrins, and PPIX standards (Frontier Scientific, Salt Lake City, UT, USA) spiking, and also through the respective PDA absorption spectra. For iron coproporphyrin (coproheme), 20 µL of about 10 mg/mL sample homogenate was mixed with 380 µL of saturated oxalic acid and heated in a boiling water bath for 1 h, and the demetallated coproporphyrin was analyzed by UPLC. The heme standard was obtained from Sigma, and the intermediate porphyrins, uroporphyrin, heptacarboxylporphyrin, hexacarboxylporphyrin, pentacarboxylporphyrin, coproporphyrin, PPIX, and Fe(III) coproporphyrin III chloride were obtained from Frontier Specialty Chemicals (Logan, UT, USA).

### 2.9. Spectral Analysis of Hemes A, B, and C

An Agilent 8453 diode array spectrophotometer (Agilent, Santa Clara, CA, USA) was used for the spectrophotometric analysis of hemes using the method by Berry and Trumpower [27] with some modifications. About 35 µL of sample was added to 1000 µL of an aqueous solution made by mixing 6 mL of pyridine, 3 mL of 1N NaOH, and 19 mL of water. The mixture was oxidized with 18 µL of 15 mM potassium ferricyanide and the stable spectrum (reaction completed) was determined. Then, some sodium dithionate (hydrosulfite) powder, about 10 mg, was mixed in to reduce the hemes and this stable spectrum was also recorded. The absorbance values at 540, 549, 558, 588, and 620 nm for reduced hemes were subtracted from the corresponding oxidized values at the same wavelengths, and the amount of heme was calculated therefrom.

### 2.10. Assay to Determine the Impact of Inhibitors on Heme Biosynthesis

In order to determine the effect of screen hits on heme biosynthesis, bacteria were grown to log phase (OD 600 of 0.2) by incubating at 30 °C, 0.6 µM prodigiosin or butylcycloheptylprodiginine was added and incubated for 3 more hours, and then, bacteria were pelleted. PBS-washed bacteria were sonicated to lyse, then extracted and assayed as described above (using spectral hemochrome analysis to measure heme and using UPLC analysis for porphyrins). Heme and porphyrin reaction intermediate levels were normalized to protein concentration. Each sample was run in triplicate, sample *n* = 1.

### 2.11. In Silico Ligand–Protein Binding Analysis

In silico docking of BCHP and prodigiosin to ChdC protein, in the coproheme binding site, was performed by staff at the University of Utah Therapeutic Accelerator, using ICM-Pro (Molsoft). Chemical structures of prodigiosin and butylcycloheptylprodiginine were obtained from PubChem. Coproheme interaction sites within the *Listeria monocytogenes* ChdC (LmChdC) complex with coproheme (X-ray crystal structure PBD ID: 5LOQ [36], B chain) served as the model for Internal Coordinate Mechanics (Molsoft ICM 64 Pro version 3.9-2b, Molsoft LLC, San Diego, CA, USA) software. Binding pocket boundaries were determined by the ICM internal coordinate force field algorithm by first considering the original ligand’s position, and then creating a 3-D force field grid centered on those coordinates that was dynamically sized to include the entire electrostatic pocket region. The box or mesh dimensions and relative position for this study used the algorithm-determined sizes. We employed the ICM Pocket Finder (SiteMap) method to identify the ligand binding sites based on the grid potential map of van der Waals interaction of key amino acid residue cavities and clefts. The position and size of the ligand binding pocket of coproheme was estimated based on Lennard–Jones potential with a Gaussian kernel, a grid map of a binding potential, and the construction of equipotential surfaces along the maps. The identified pockets were represented as a surface, and the dimensions of each pocket included the pocket volume, area, solvent exposure, and hydrophobicity. These properties were calculated using ICM Pocket Finder and quantified as a Drug-Like Density (DLID) score. An ICM-Pro scoring function also produces a score that approximates binding free energy between the ligand and protein. The lower the ICM score, the higher the chance that the ligand can bind. This score was calculated for docked ligands.

## 3. Results and Discussion

### 3.1. Creation and Validation of Bacterial Strains Used for Competitive Bacterial Viability Assay

We developed a competitive bacterial viability assay to screen for drugs that selectively inhibit monoderm-specific CPD pathway enzymes [24]. This was accomplished by replacing the canonical PPD pathway genes in an (diderm) *E. coli* strain with CPD pathway versions [16,17] from *S. aureus*. The growth of treated wild-type and replacement strains in mixed culture, with and without heme pathway rescue, was followed by measuring their strain-specific fluorescence. Two versions of *E. coli* MC4100 expressing either a YFP or CFP fluorophore, genetically identical at all other loci, were used in the development of our screen. The CFP version (MC4100-CFP or “WT-CFP”) was used as the wild-type control strain in our screen, utilizing the endogenous PPD pathway [37] for the biosynthesis of heme. We modified the YFP-expressing strain (MC4100-YFP) by deleting the three terminal PPD pathway genes (*pgdH1*, *cgdC*, and *ppfC*) and replacing them with the corresponding genes from *S. aureus* (*cgoX*, *cpfC*, and *chdC*) (Figure 4) [38].

The competition viability assay has an advantage over a simple viability assay, as the competition viability assay (mixed cultures) is not as sensitive to the time of assay compared to cultures of individual bacterial strains. In a competition assay, inhibited bacteria will be outcompeted by their non-inhibited counterparts, which will deplete essential media components, simplifying the endpoint, whereas sensitive bacteria growing individually may continue to grow if they have only been partially inhibited at that concentration.

Although the triple heme gene replacement mutants possess a functional biosynthesis pathway for heme, we observed that their growth was not as robust as wild type. Altered flow through the biosynthetic pathway could lead to the buildup of toxic reaction intermediates. Heme and porphyrin analyses revealed that protoporphyrin IX builds up in the triple replacement mutant (Table 1). Since the CPD pathway enzymes do not generate protoporphyrin IX, it was clear that some residual *E. coli*-specific coproporphyrinogen decarboxylase activity must still remain. *E. coli* has a second enzyme encoded by *CgdH*, which catalyzes the same decarboxylation step (anaerobically) as CgdC. If the CgdH-catalyzed coproporphyrinogen III-to-protoporphyrinogen IX decarboxylation step occurs within a remnant nonfunctional diderm heme biosynthesis pathway, toxic intermediates may build up. Protoporphyrin IX buildup can be particularly toxic. For this reason, we knocked out *cgdH*, creating the strain used in our screen “*Sa*-CPD-YFP”. An assay of the porphyrins protoporphyrin IX, uroporphyrin, heptacarboxylporphyrin, hexacarboxylporphyrin, pentacarboxylporphyrin, and coproporphyrin showed that *cgdH* KO dramatically reduced PPIX buildup in the replacement mutant strain (Table 1). The total porphyrins remained elevated in the mutant. The profiles of individual porphyrins (percent of the total) generally resemble that of wild type (approximately 25% coproporphyrin), with a small elevation in partially decarboxylated forms (Table 1). The peak for coprohemin (iron coproporphyrin) was very small in mutant bacteria. Because our porphyrin assay oxidizes porphyrinogens to porphyrins prior to UPLC analysis, we do not know whether CgoX or CpfC has become the bottleneck. It is likely that the reaction catalyzed by CgoX will occur spontaneously, albeit at a slower rate, so we expect that *S. aureus* CpfC is the bottleneck in our mutant strain of *E. coli*. The *cgdH* knockout noticeably increased the robustness of our screening strain.

Gene replacements were verified by qPCR analysis and by sequencing of the introduced genes and surrounding areas to rule out unintended changes that may have occurred during the homologous recombination process. One coding variant was observed in chdC (compared to the predicted) (c.79C > A), which would result in a coding change (Leu-27-Ile). We believe that this is likely a natural variant due to multiple sequenced clones having the same sequence at this location. qPCR analysis demonstrated that the introduced *S. aureus* genes were expressed in *Sa*-CPD-YFP, and that the replaced *E. coli* genes were no longer expressed in *Sa*-CPD-YFP (Appendix A).

In the construction of the triple replacement (i.e., *cgoX* + *cpfC* + *chdC*) it was essential to grow each intermediate strain on media containing heme until all three genes were replaced. This confirms that without a complete biosynthetic pathway the strain remains auxotrophic for heme and terminal biosynthetic intermediates are not interchangeable.

### 3.2. The Screen-Competitive Bacterial Viability Assay Successfully Identified Specific Hits

In our screen assay, a stoichiometric mixture of *Sa*-CPD-YFP (expressing YFP) and wild-type bacteria strains (expressing CFP) wsa incubated with potential inhibitors, and then, YFP and CFP fluorescence readings were taken to quantify bacterial growth, normalizing to the respective fluorescence signals in vehicle-treated control wells. Matching heme–pathway–rescue control plates were also plated containing hemin (the product of the targeted pathway) to facilitate the selection of inhibitors that specifically target the heme pathway and filter out false positives that diminish growth by other means. The well readings for two screen hits, butylcycloheptylprodiginine (BCHP) and prodigiosin, are shown in Table 2 and Appendix A. BCHP specifically reduced the *Sa*-CPD-YFP bacteria to 4% of the vehicle control at 16 h. When *Sa*-CPD-YFP growth was suppressed, the final amount of WT-CFP bacteria in the mixed culture increased (149% of the control) without competition from *Sa*-CPD-YFP. Upon hemin addition, *Sa*-CPD-YFP and WT-CFP returned to near-normal levels of 70% and 99%, respectively.

Bacteria are inoculated at a concentration that allows for logarithmic growth to occur during the 16 h incubation. If the drug kills or suppresses growth of one strain over the other, there is a dramatic and reproducible difference in the fluorescent signal between the two strains. It is also typical to see an increase in the growth yield of the wild-type strain when *Sa*-CPD-YFP growth is solely inhibited, another indication of specificity (Appendix A). This additional increase in WT-CFP bacteria is due to decreased competition for the nutrients available in the media. The fact that these genetically matched bacteria can be monitored in competition allows normalization for plate-to-plate environmental variation and controls for off-target inhibition. Calculated Z’ factors for the primary screen reflect its robust reproducibility. The readings from an assay plate used to calculate the Z’ factor for our screen are shown in Appendix A. The Z’ factor for the *Sa*-CPD-YFP YFP channel (ex/em = 500/536 nm) is 0.76 and WT-CFP CFP channel (ex/em = 445/475) is 0.79. Z’ factors above 0.5 are considered excellent [33].

Hits from our initial screens include several series of related compounds, such as those shown in Figure 6, demonstrating the reliability of the screen. Butylcycloheptylprodiginine and prodigiosin (Figure 6D,E, Appendix A) both belong to the same family of bioactive bacterial metabolites. These natural products have been shown to have antibacterial activity, and the observation has been made that prodigiosin has a greater inhibitory effect on monoderm bacteria such as *S. aureus* than on diderm bacteria such as *E. coli* [39]. Prodigiosin has also been studied for its ability to trigger apoptosis in malignant cancer cells, but it has been reported to be non-genotoxic, as well as non-toxic in epithelial cells [40]. There are a varied range of biological effects that have been attributed to this family of molecules; however, the actual mechanism of action is not clear. It is interesting that these prodigiosin derivatives share chemical similarities with heme and with heme biosynthetic intermediates. Heme is a tetrapyrrole, while the prodigiosin derivatives are tripyrroles. It is easily conceivable that these molecules interact with the substrate-binding pocket of a heme biosynthesis enzyme.

Miconazole nitrate (MN) is one of the azole compounds identified in our screen of the MicroSource Spectrum Collection, from the group B similarity group in Figure 6B. It did not inhibit the wild-type bacteria in our screen at the concentrations tested, but it did inhibit the *Sa*-CPD-YFP strain, and this growth inhibition was reversed when heme was added. These data suggest that the growth limitation in these bacteria is related to their inability to synthesize heme and suggests that one of the three replacement enzymes in the *Sa*-CPD-YFP strain is the primary target. Miconazole nitrate is used as an antifungal. It is known to interact with the heme iron in a yeast cytochrome P450 enzyme (CYP51/Erg11p), which decreases the synthesis of a critical membrane sterol in the fungi, as well as inhibiting peroxidase and catalase activities (hemoproteins) at high concentrations, leading to cellular damage. It has been shown that the azole drugs have activity against *M. tuberculosis* [41]. Azole drugs have been proposed to also coordinate with the heme iron of *M. tuberculosis* P450 enzymes. The sequencing of all P450-encoding genes in an azole-resistant mutant of *M. bovis* did not reveal any P450 mutations that would support this hypothesis [41,42]. These data call into question whether the P450 enzymes are the true targets for the azole drugs in *Mycobacterium*.

Interestingly, the general three-ringed structure found in the Group A backbone shown in Figure 6 is found in Clofazimine and Riminophenazine, compounds currently in use or under development for the treatment of tuberculosis. Although the mechanisms of action of Clofazimine are not yet entirely understood, suggested targets include the outer membrane, the respiratory chain, and ion transporters [43]. Riminophenazine, in Phase 1 clinical development, is reported to inhibit ion transport and bacterial respiration [44].

### 3.3. Analysis of Screen Hits by Titration in a Competitive Viability Assay Identified Two Strong Inhibitors

Hits that met our criteria of the selective inhibition of the *Sa*-CPD-YFP strain and not the wild-type bacteria, with inhibition that was rescued by hemin, were further analyzed by titrating drug concentrations in the competitive viability assay used for the screen (Figure 7 and Appendix A). Serial five-fold dilutions of compounds were generated and used to treat mixtures of wild-type and mutant strains either in the absence or presence of hemin, using screen assay conditions. Two strong hits were identified in the NCI Natural Products Set IV Library, prodigiosin (NSC# 47147) and butylcycloheptylprodiginine hydrochloride (BCHP) (NSC# 247562). Both prodigiosin and BCHP exhibited nanomolar IC50s in the competitive viability assay (Appendix A). Prodigiosin and BCHP, both prodiginines, have the same parent nucleus but different side groups. We purchased and tested two other prodiginines, obatoclax and undecylprodigiosin. Inhibition was also observed with these molecules (Figure 7). These four prodiginines were also titrated into cultures of *Sa*-CPD-YFP alone (no WT-CFP competitor) (Appendix A), using primary screen conditions, and inhibition was observed. Another chromogenic bacterial secondary metabolite, Violacein, also studied for its antimicrobial properties, was purchased and did not show activity in our screen (below 5 µM).

### 3.4. Prodigiosin and BCHP Inhibition Is Not an Artifact of Fluorescence Interference

Prodigiosin and BCHP are both colored compounds. In order to test whether the observed inhibition was an artifact of compound spectral properties, additional controls were run. The compound was also titrated into dedicated control wells and mixed after the cells had grown for 18 h at 30 °C in the shaking incubator in 96-well plates, shortly before taking fluorescent readings (Figure 8). The data show that while prodigiosin and BCHP interfered with CFP fluorescence at high concentrations, they did not have an experimentally significant effect at lower concentrations. We conclude that prodigiosin and BCHP interfere with fluorescence above 5 µM, but they were not responsible for the low nM inhibition observed in the competition viability assay.

### 3.5. Heme and Porphyrin Analysis

Since the inhibition of heme biosynthesis will decrease cellular heme levels in growing bacteria, it presents an additional way to validate hits for this screen. A preliminary spectral analysis of heme content revealed that the heme levels decreased by 1.6- and 5.8-fold with prodigiosin and butylcycloheptylprodiginine treatment, respectively (Figure 9). UPLC analysis of intermediate porphyrins showed increased levels of Fe-coproporphyrin (coproheme) in treated *Sa*-CPD-YFP bacteria relative to control. This would be consistent with the inhibition of *S. aureus* ChdC. Preliminary UPLC porphyrin analysis detected increased Fe-coproporphyrin levels in Miconazole Nitrate (MN)-treated bacteria, suggesting that MN may also function by inhibiting ChdC (Appendix A). Spectral hemochrome analysis also showed decreased heme levels upon MN treatment compared to controls, but the levels for the drug-treated sample were below detection. 

### 3.6. Measured Antimicrobial Activity against Cultures of S. aureus Newman

Our screen tested the antimicrobial activity of hits using diderm strains (of *E. coli*) that are genetically modified to carry heme genes from a monoderm (*S. aureus)*. However, the “real” monoderm targets of the screen, Actinobacteria and Firmicutes, have a different cell wall composition and do not have the outer LPS-containing membrane found in *E. coli*. Because of the outer membrane barrier and a plethora of MDR pumps, diderms are considered more difficult to target compared to Firmicutes as they are much more efficient at keeping out drugs. Because of these differences, we tested for activity in strains of Firmicutes. To test for inhibition, five low-µM hits from the MicroSource Spectrum Collection were examined for their ability to inhibit the growth of *S. aureus* in liquid culture. *S. aureus* Newman cultures were grown in rich medium containing hit compounds at 0, 2, and 10 µM concentrations. The absorbance at 600 nm was monitored for 5 h. We determined that three of the five were effective inhibitors when tested at 2 and 10 µM (Table 3).

### 3.7. In Silico Docking of Butylcycloheptylprodiginine and Prodigiosin with a ChdC Protein Structure Is Consistent with the Hypothesis That These Molecules Inhibit by Binding to the ChdC Active Site

Chemical structures for the small molecules BCHP and prodigiosin were docked against the molecular coordinates of *Listeria monocytogenes* coproheme decarboxylase (LmChdC) (X-ray crystal structure PBD ID: 5LOQ) in the coproheme binding site. Analysis revealed that there is reasonable space within the substrate binding site of ChdC to accommodate these prodiginine molecules. ICM-Pro produces a score that approximates the binding free energy between the ligand and protein. The lower the ICM score, the higher the chance that the ligand can bind. Docked BCHP and prodigiosin ligands yielded strong ICM scores, predicting binding interactions between ChdC and BCHP (−38.18) and between ChdC and prodigiosin (−50.18). This supports the hypothesis that these molecules could inhibit by competitively binding to the ChdC active site. The most favorable docking positions for prodigiosin are shown in Figure 10, superimposed with the coproheme ligand from the structure. Appendix A shows the docking for BCHP. The docking poses show good correspondence between the atoms of the docked BCHP and prodigiosin and the original ligand.

## 4. Conclusions

Above, we detailed the development of an efficient, robotic, scalable growth assay-based screen to identify compounds that possess antimicrobial activity against monoderm bacteria in the Firmicute species (Figure 11). This screen identifies compounds that inhibit these bacteria by targeting monoderm-specific heme biosynthetic (CPD) pathway enzymes, ChdC, CpfC, or CgoX. An additional value of this screen is that only compounds that are taken up by live bacteria will be identified. This is in contrast to in silico-based drug design or in vitro enzyme assay-based screens that do not distinguish compounds that may not be functional in vivo. As a proof of concept, we screened small, available libraries and identified compounds that specifically inhibited the bacterial growth of bacteria possessing the CPD pathway. While certainly not exhaustive, the proof of concept that compounds can be identified using an in vivo screen is very attractive. An analysis of heme pathway intermediate accumulation by bacteria exposed to butylcycloheptylprodiginine and miconazole nitrate revealed that both had accumulated coproheme. This is consistent with the in vivo inhibition of the CPD-specific enzyme ChdC.

## Figures and Tables

**Figure 1 biomolecules-13-01485-f001:**
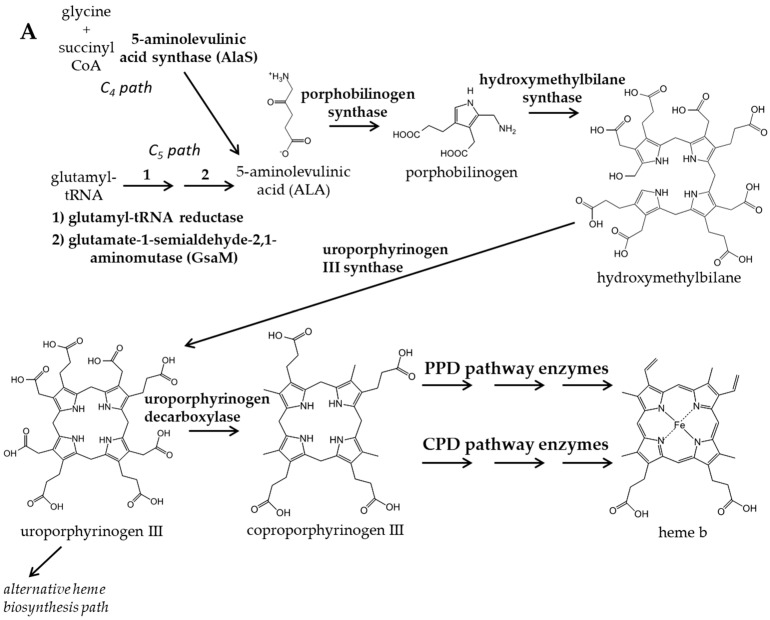
An overview of the heme biosynthesis pathway is depicted in (**A**). PPD and CPD pathways diverge in the last three enzymatic steps, which are detailed in (**B**). The last three heme biosynthesis steps are conserved in mammals and diderm bacteria; however, the order of oxidation of the macrocycle, insertion of the iron, and decarboxylation differ in the monoderm bacteria. In (**B**), enzymes that catalyze similar biosynthetic steps are colored similarly, as are associated protein names: orange for decarboxylation step, green for oxidation of the macrocycle, and blue for iron insertion. Orange circles over intermediates indicate chemical groups modified from previous step. No circles appear on heme because previous steps diverge. When assaying for intermediates by UPLC, porphyrinogens are first oxidized to porphyrins. Here, we describe replacing *E. coli* PPD genes with *S. aureus* CPD genes in *E. coli* bacteria that have fluorescent tags integrated into the genome; this fluorescence is used to follow growth and/or growth inhibition as a screen to identify specific inhibitors of Firmicute bacteria.

**Figure 2 biomolecules-13-01485-f002:**
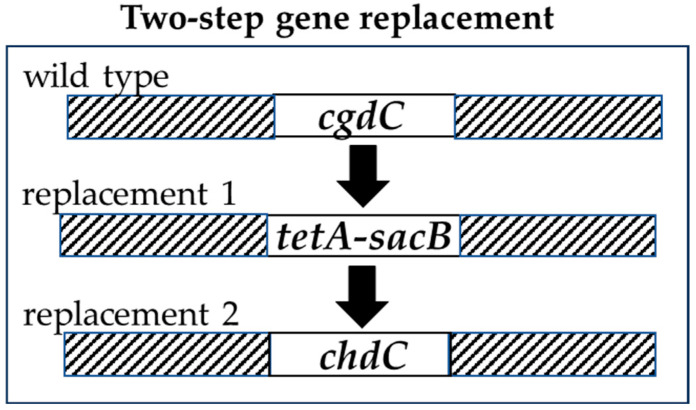
Selected *E.coli* heme genes were replaced using a two-step strategy that allowed for selection of homologous recombinants. First, the endogenous heme gene was replaced with a cassette that allows selection (tetracycline resistance, sucrose, and fusaric acid sensitivity). Second, the cassette was replaced with an exogenous heme gene.

**Figure 3 biomolecules-13-01485-f003:**
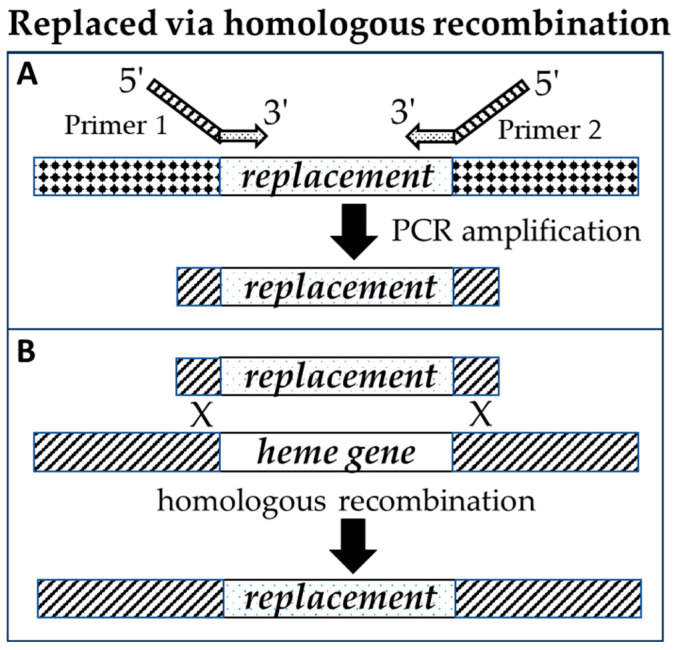
In our gene replacement strategy, (**A**) new inserts were amplified using chimeric primers composed of regions complementary to the amplicon, and with regions of homology to the desired integration site in the target genome; (**B**) gene replacements occurred through homologous recombination.

**Figure 4 biomolecules-13-01485-f004:**
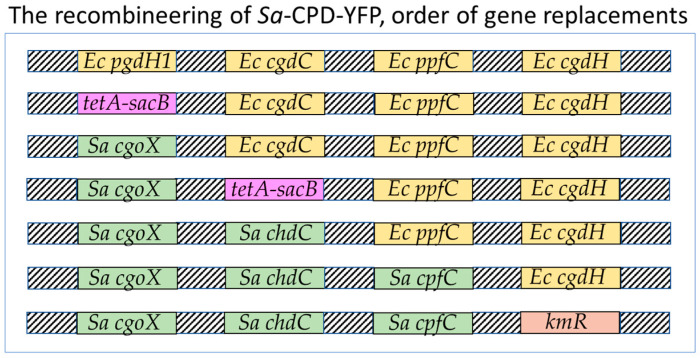
The recombineering of *Sa*-CPD-YFP. *E. coli* “*Ec*” heme genes were replaced with *S. aureus* “*Sa*” heme genes in the order shown (top to bottom). *cgdH* was replaced with a kanamycin resistance marker, kmR.

**Figure 5 biomolecules-13-01485-f005:**
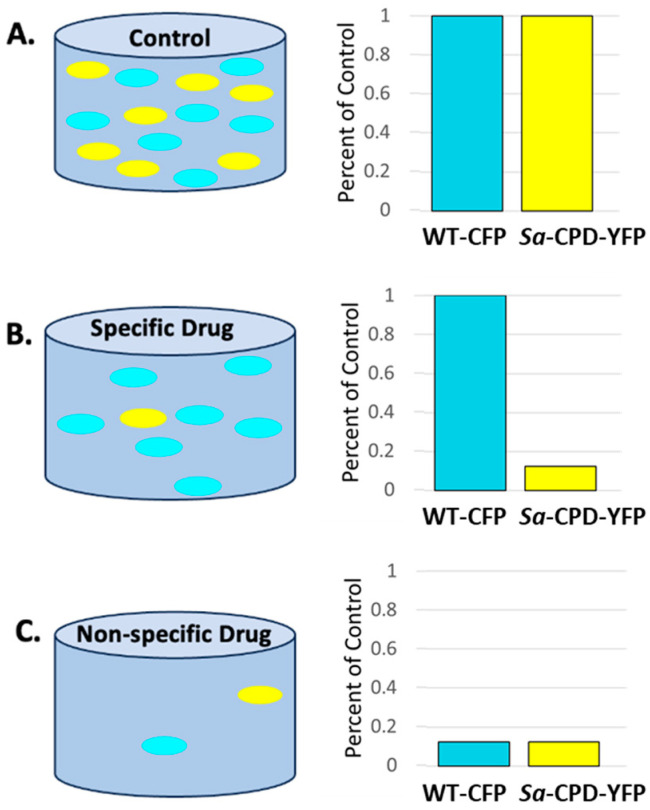
WT-CFP, *E. coli* control bacteria (PPD heme genes) with cyan fluorescence and *Sa*-CPD-YFP, *E. coli* with CPD heme genes and yellow fluorescence were mixed in our assay. Fluorescence was normalized to vehicle-treated control at both wavelengths. Inhibitors specifically targeting CPD or PPD heme biosynthesis enzymes should result in a decrease in one fluorescent signal without change in the other. Nonspecific inhibitors will impact the growth of both strains similarly. Here, WT-CFP and *Sa*-CPD-YFP bacteria are represented as blue and yellow ovals, respectively, in wells (grey), and represent scenarios where no (**A**), one (**B**), or both (**C**) strains are inhibited by drug. To the right of each illustration are plots representing the relative fluorescence signal intensities of each bacterium, normalized to untreated wells.

**Figure 6 biomolecules-13-01485-f006:**
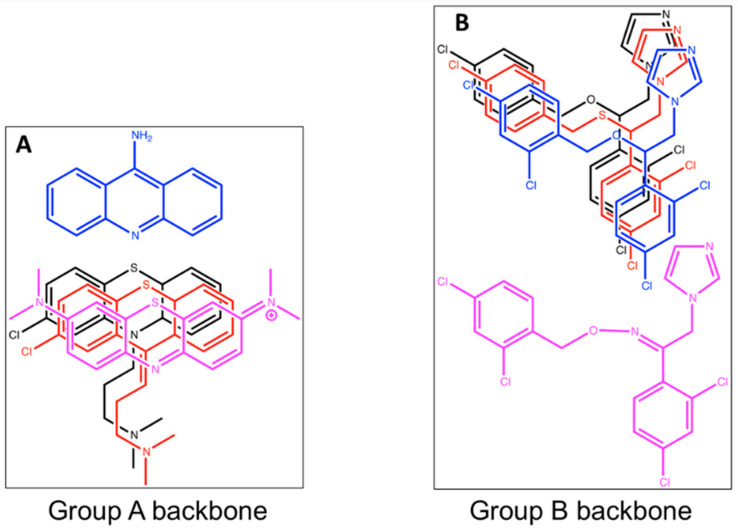
Compounds from the initial screen showing overall similar architecture within multiple hits, and heme B. Eight potential hits from a MicroSource Spectrum library screen (**A**,**B**) exhibit strong chemical similarity to other hits in the same screen. Structures of these compounds are shown. Acrisorcin is the primary member of Group A (**A**). Miconazole nitrate is the primary member of Group B (**B**). Some Group B members are marketed as antifungal drugs with associated antibacterial properties. The two strongest hits from our small-scale screens butylcycloheptylprodiginine (CID:135539743) (**D**) and prodigiosin (CID:135455579) (**E**) are tripyrroles; (**C**) Heme B (CID:26945) is a tetrapyrrole. The substrates of CpfC and ChdC are relatively planar porphyrins. CgoX has a more flexible porphyrinogen as a substrate. PubChem was used to retrieve the CID from the National Center for Biotechnology Information 15 August 2023.

**Figure 7 biomolecules-13-01485-f007:**
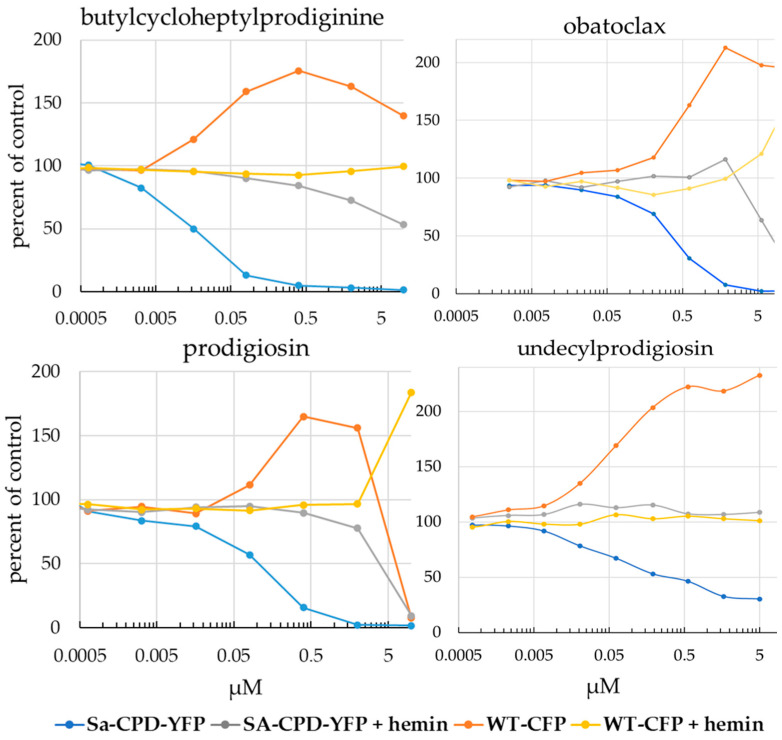
Serial five-fold dilutions of compound were generated and used to treat mixtures of *Sa*-CPD-YFP and WT-CFP bacteria, either in the absence or presence of hemin, using assay conditions similar to those in the pilot screen. Cultures were measured at 16–18 h in both YFP (*Sa*-CPD-YFP) and CFP (WT-CFP) channels. Prodiginine family compounds specifically inhibit *Sa*-CPD-YFP. When hemin was added to cultures in a second assay plate to bypass any block in heme biosynthesis, the inhibition of the *Sa*-CPD-YFP strain was not observed, indicating that drug is targeting one of the three CPD pathway enzymes.

**Figure 8 biomolecules-13-01485-f008:**
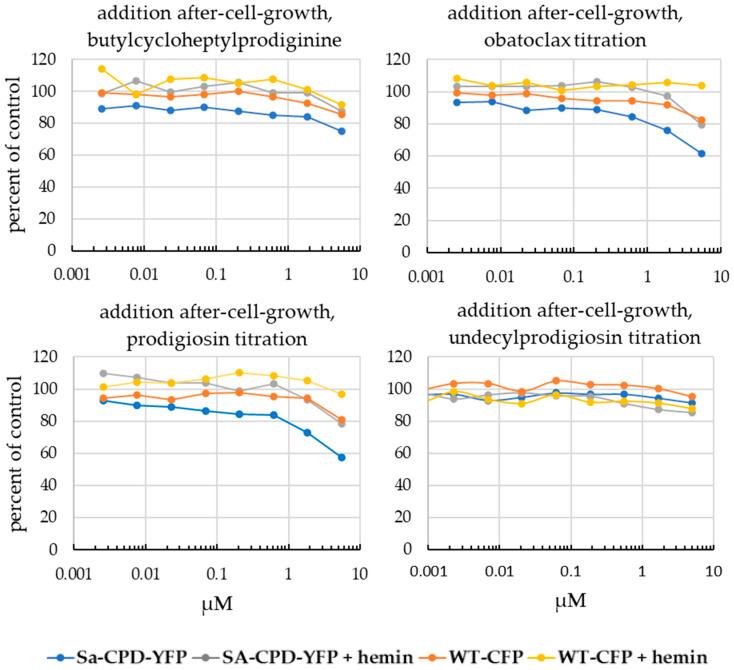
In order to test whether observed inhibition was an artifact of prodiginine’s spectral properties, compounds were also titrated into dedicated control wells and mixed after the cells had grown for 18 h under screen conditions, shortly before taking fluorescent readings in both the YFP (*Sa*-CPD-YFP) and CFP (WT-CFP) channels.

**Figure 9 biomolecules-13-01485-f009:**
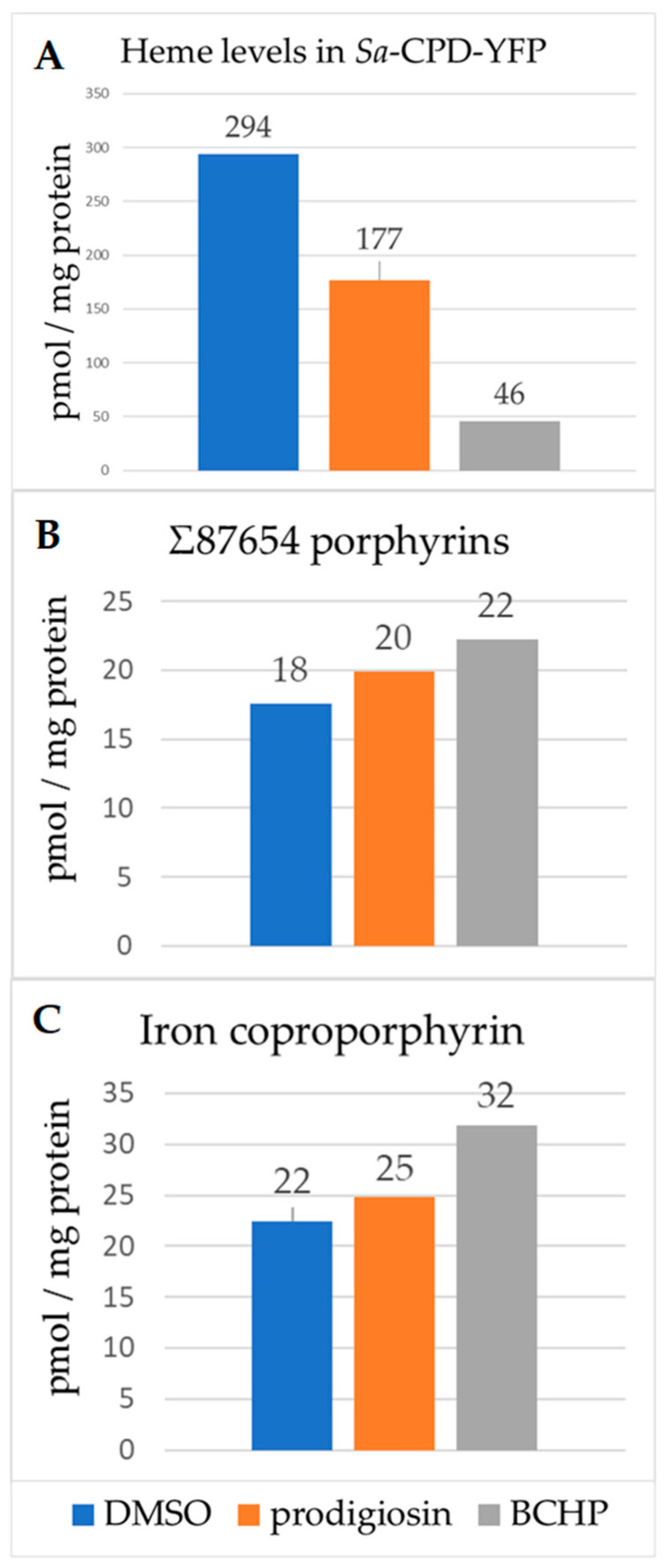
Tetrapyrrole and heme analysis in treated *Sa*-CPD-YFP bacteria. Bacteria were grown at 29 °C until O.D._600_ of 0.2; compounds were added at 0.6 µM and incubated for an additional 3 h. Cells were then pelleted, washed, and subjected to spectral hemochrome analysis (heme) or UPLC analysis (porphyrins). (**A**) Heme levels decreased by 1.6- and 5.8-fold with prodigiosin and butylcycloheptylprodiginine, respectively. There was a corresponding increase in Σ87654 porphyrin levels (intermediate porphyrins with 8, 7, 6, 5, and 4 carboxyl groups) (**B**) and Fe-coproporphyrin levels (**C**). Each sample was run in triplicate, sample *n* = 1.

**Figure 10 biomolecules-13-01485-f010:**
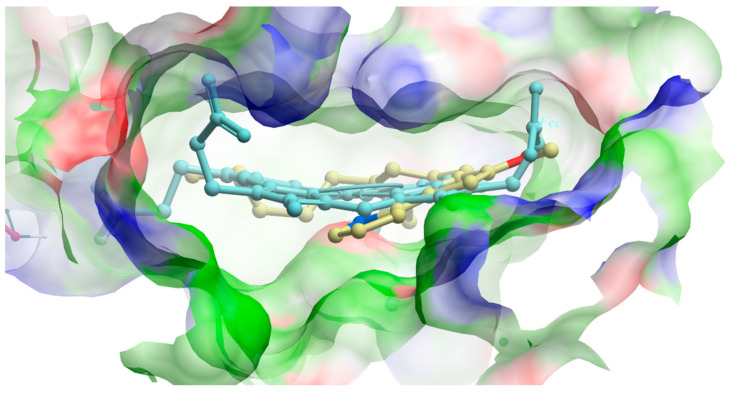
Results of in silico docking of prodigiosin with LmChdC, PDB ID: 5LOQ. Docked prodigiosin is shown in the center of this figure as a thick ball and stick model (carbon yellow, oxygen red, and nitrogen blue), superimposed with the coproheme ligand from the 5LOQ structure (light blue carbons and sticks). The surrounding ligand binding pocket surface is colored by binding property (hydrophobic areas in green, hydrogen bond acceptors in red, and hydrogen bond donors in blue). There is reasonable space within the binding pocket to accommodate prodiginine molecules.

**Figure 11 biomolecules-13-01485-f011:**
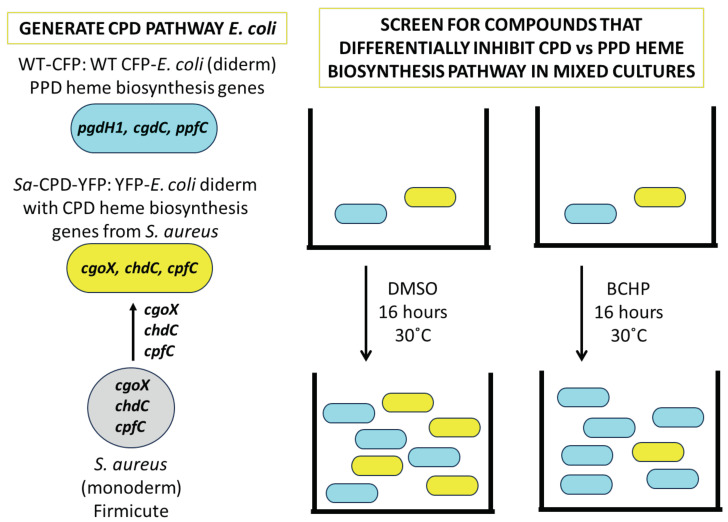
Yellow fluorescent protein (YFP) expressing *E. coli* (diderm) bacteria were modified to contain CPD heme biosynthesis genes from the monoderm Firmicute bacteria *S. aureus* (“*Sa*-CPD-YFP”), replacing PPD heme biosynthesis genes. These modified bacteria were mixed with matching wild-type *E. coli* expressing cyan fluorescent protein (CFP) in a screen to identify small molecules that differentially inhibit the CPD vs. PPD heme biosynthesis pathway. The PPD pathway is found in humans as well as diderms, whereas the CPD pathway is not. This screen facilitates the identification of antimicrobials that are specific to Firmicute bacteria. The example inhibitor shown, BCHP, is butylcycloheptylprodiginine.

**Table 1 biomolecules-13-01485-t001:** Heme and porphyrin analysis of parental and intermediate heme gene replacement strains to analyze the effect of *cgdH* knockout. PPIX is protoporphyrin IX. Porphyrin Σ87654 is total intermediate-extracted porphyrins with 4, 5, 6, 7, or 8 carboxyl groups. All quantities are in pmol/mg sample protein.

	Hemin	PPIX	Porphyrin Σ87654	Uro	Hepta	Hexa	Penta	Copro
*E. coli* Strain	pmol/mg protein	pmol/mg protein	pmol/mg protein	% mol	% mol	% mol	% mol	% mol
parental MC4100-YFP	463.5	0.4	0.4	74.9	1.2	0	0	23.9
*S. aureus* triple mutant *	48.5	68.6	265.8	58.1	5.6	1.7	7.6	27
*S. aureus* triple mutant plus *cgdH* KO **^‡^**	46.6	3.7	449.3	59.3	7.0	1.9	6.8	25.1

* lab name SYHQ1: *pgdH1*, *cgdC*, and *ppfC* genes replaced with *S. aureus cgoX*, *chdC*, and *cpfC*. **^‡^** lab name SYHQ-NKAN1: SYHQ1 with *cgdH* replaced with a kanamycin resistance marker.

**Table 2 biomolecules-13-01485-t002:** Microplate readings (percent of control) in CFP and YFP channels after 16 h incubation with BCHP or prodigiosin, for duplicate wells +/− heme. Excitation/emission: YFP 500/535, CFP 445/475.

Treatment	*Sa*-CPD-YFP	WT-CFP	*Sa*-CPD-YFP + Hemin	WT-CFP + Hemin
BCHP	4	149	70	99
prodigiosin	2	46	73	110

**Table 3 biomolecules-13-01485-t003:** Toxicity testing against *S. aureus.* Percent growth inhibition measured at 5 h in rich LB medium.

Compound	Inhibition 0 µM	Inhibition 2 µM	Inhibition 10 µM
Chlorprothixene	0	0	0
Diphenyl urea	0	0	0
Benzethonium	0	76%	93%
Clotrimazole	0	40%	70%
Miconazole	0	79%	83%

## Data Availability

The data will be made available from the CCEH.io website and strains will be made available once an institutional MTA is signed with the University of Utah.

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
