# Peer review of "Exploiting Differences in Heme Biosynthesis between Bacterial Species to Screen for Novel Antimicrobials"

_biomolecules, 2023, doi:10.3390/biom13101485_

Round 1

Reviewer 1 Report

The manuscript "Screening for Anit-Microbials Targeting A Novel Pathway in Heme Biosynthesis" is a well-written research paper, dealing with a very important topic. The screening approach to target enzymes involved in the coproporphyrin heme biosynthesis pathway in order to find novel lead substances for future antibiotics is designed in an elegant and efficient way and convincingly presented. Also first inhibiting substances are presented. I especially enjoyed the detailed and precise methods section and also the results are presented clearly and are easy to follow. The emphasis on all relevant controls is also necessary and is presented nicely. I am sure that this work will be of great help in the future, as it delivers the basis for translational research in order to develop novel antibiotic therapeutics targeting the heme biosynthesis pathway of Gram-positives.

Some minor issues need to be clarified:

- in the abstract: in the first sentence it reads: "... between di- and monoderm bacteria. The latter employs the protoporphyrin-dependent (PPD) pathway, while the former utilizes the more recently discovered...." This should be the other way round; diderm utilize the PPD, monoderm utilize the CPD pathway

- in the abstract: the second last sentence talks about the enzyme ChdC, which is referred to coproheme dehydrogenase instead of coproheme decarboxylase (later in the manuscript it is correctly addressed)

- In the first sentence of the introduction it is mentioned that heme is important for many biological processes and then referred to as ferrous-protoporphyrin IX. Which is not giving the complete picture, as in many enzymes, also in the listed ones (e.g. peroxidases, catalases,...) it is present as ferric-protoporphyrin IX.

- In the introdcution in the section, in which the small colony variant of S. aureus is discussed, the authors should also cite the following work, which reports a small colony variant depending on deletion of chdc. (https://pubmed.ncbi.nlm.nih.gov/23737523/)

- In the introduction: the second last sentence of the first paragraph and the first sentence of the last paragraph seem to be repetitive (...were believed to be conserved among prokaryotes and metazoans.../ ...accepted belief was that the heme biosnthetic pathway was conserved among all organisms...)

- Figure 1: The figure legend is insufficient. Colours of enzyme names need to be explained. Also even though ChdC and AhbD have the same substrate and same product, the mode of function is completely different and AhbD is part of the siroheme dependent pathway. To place these to enzymes next to each other in this figure is misleading.

- In the second last paragraph of the introduction: citations should be given in the section about the "newly identified enzyme, coproheme decarboxylase,...."

- In section 3.7.: Why did the authors perform docking on 5LOQ, with bound co-factor. There is a structure of apo-LmChdC also available (4WWS). In the methods it is described that the interaction sites were analyzed from 5LOQ, but docking itself could have also been performed using 4WWS.

- In the conclusions: In the first sentence it is stated that compounds were identified that possess antimicrobial activity specifically against Firmicute and Actinobacteria. While this might be true, in this work the terminal CPD enzymes from Staphylococcus aureus (Firmicute) were investigated and docking was performed on the ChdC of Listeria monocytogenes (Firmicute). There are well known structural and functional differences in actinobacterial ChdCs (a catalytically important distal histidine is lacking in firmicute ChdCs, https://pubmed.ncbi.nlm.nih.gov/32440366/). Potentially the identified substances interact differently with actinobacterial enzymes.

Author Response

Reviewer #1

The manuscript "Screening for Anit-Microbials Targeting A Novel Pathway in Heme Biosynthesis" is a well-written research paper, dealing with a very important topic. The screening approach to target enzymes involved in the coproporphyrin heme biosynthesis pathway in order to find novel lead substances for future antibiotics is designed in an elegant and efficient way and convincingly presented. Also first inhibiting substances are presented. I especially enjoyed the detailed and precise methods section and also the results are presented clearly and are easy to follow. The emphasis on all relevant controls is also necessary and is presented nicely. I am sure that this work will be of great help in the future, as it delivers the basis for translational research in order to develop novel antibiotic therapeutics targeting the heme biosynthesis pathway of Gram-positives.

Some minor issues need to be clarified:

- in the abstract: in the first sentence it reads: "... between di- and monoderm bacteria. The latter employs the protoporphyrin-dependent (PPD) pathway, while the former utilizes the more recently discovered...." This should be the other way round; diderm utilize the PPD, monoderm utilize the CPD pathway

Thank you for this comment.  This has now been fixed.

- in the abstract: the second last sentence talks about the enzyme ChdC, which is referred to coproheme dehydrogenase instead of coproheme decarboxylase (later in the manuscript it is correctly addressed)

This has now been fixed.

- In the first sentence of the introduction it is mentioned that heme is important for many biological processes and then referred to as ferrous-protoporphyrin IX. Which is not giving the complete picture, as in many enzymes, also in the listed ones (e.g. peroxidases, catalases,...) it is present as ferric-protoporphyrin IX.

Well, both are correct since Fe-PPIX is heme and heme can be Fe2+ or Fe3+ so changed to:

“Many biological processes depend upon heme, an iron-containing porphyrin (iron protoporphyrin IX) as a cofactor in part because of the ability of the iron to be in the 2+ or the 3+ state.”

- In the introdcution in the section, in which the small colony variant of S. aureus is discussed, the authors should also cite the following work, which reports a small colony variant depending on deletion of chdc. (https://pubmed.ncbi.nlm.nih.gov/23737523/)

This reference has been added.

- In the introduction: the second last sentence of the first paragraph and the first sentence of the last paragraph seem to be repetitive (...were believed to be conserved among prokaryotes and metazoans.../ ...accepted belief was that the heme biosnthetic pathway was conserved among all organisms...)

The repetitive language here has been removed.

The classical heme biosynthetic pathway in plants has been targeted and several herbicides which inhibit plant protoporphyrinogen oxidase are now widely used [8].  However, the perceived problem with targeting enzymes of the heme biosynthetic pathway to treat bacterial pathogens was that the enzymes of the pathway were believed to be conserved among prokaryotes and metazoans. Thus, target compounds toxic to the infectious agent would be equally toxic to the host. 

For over half a century the accepted belief was that the heme biosynthetic pathway was conserved among all organisms with all metabolic intermediates invariable.  This so-called “classic pathway” possesses alternative enzymes at two steps to accommodate aerobic vs. anaerobic lifestyles, i.e., decarboxylation of coproporphyrinogen III to protoporphyrinogen IX (aerobic: CgdC, anaerobic: CgdH) [9-11] and the oxidation of protoporphyrinogen IX to protoporphyrin IX  (aerobic: PgoX, mixed: PgdH1/2) [11-13] (Figure.1).

But now is:

The classical heme biosynthetic pathway in plants has been targeted and several herbicides which inhibit plant protoporphyrinogen oxidase are now widely used [8]. 

For over half a century the accepted belief was that the heme biosynthetic pathway was conserved among all organisms with all metabolic intermediates invariable.  Thus, target compounds toxic to a prokaryotic infectious agent would be equally toxic to a metazoan host.  This so-called “classic pathway” possesses alternative enzymes at two steps to accommodate aerobic vs. anaerobic lifestyles, i.e., decarboxylation of coproporphyrinogen III to protoporphyrinogen IX (aerobic: CgdC, anaerobic: CgdH) [9-11] and the oxidation of protoporphyrinogen IX to protoporphyrin IX  (aerobic: PgoX, mixed: PgdH1/2) [11-13] (Figure.1).

- Figure 1: The figure legend is insufficient. Colours of enzyme names need to be explained. Also even though ChdC and AhbD have the same substrate and same product, the mode of function is completely different and AhbD is part of the siroheme dependent pathway. To place these to enzymes next to each other in this figure is misleading.

Figure 1 has been remade to address comments of various reviewers.  Colors of enzyme names have now been explained in the figure legend.  AhbD has been removed from this figure as suggested.

- In the second last paragraph of the introduction: citations should be given in the section about the "newly identified enzyme, coproheme decarboxylase,...."

References were added to ChdC.

- In section 3.7.: Why did the authors perform docking on 5LOQ, with bound co-factor. There is a structure of apo-LmChdC also available (4WWS). In the methods it is described that the interaction sites were analyzed from 5LOQ, but docking itself could have also been performed using 4WWS.

As reviewer 1 says, images provided to us from the University of Utah therapeutic accelerator, were of our prodiginine ligands docked against the protein molecular coordinates of LmChdC from 5LOQ.  The program used “removes” bound ligand from the pocket prior to docking, creates a 3D box of the active site that includes all essential amino acids of the original pocket and additional extents.  It then docks into that space without “knowledge” of the original ligand position.  Docking uses forcefields that approximate potential binding energies at each point along a grid.  Scoring includes information about entropy loss, solvation, hydrogen bond interactions, electrostatic energy, and hydrophobic energy.  Our results show that there is adequate space in the binding pocket for the ligand, and the strong negative ICM scores suggest that the prodiginine ligands will bind well to this space.  Proteins can differ somewhat in ligand bound and apo form, but a prodiginine bound form could differ from both.  It would be interesting to look at binding to the apo form and it is a good suggestion for future studies, but we do not feel that it would add a lot to the current paper.

- In the conclusions: In the first sentence it is stated that compounds were identified that possess antimicrobial activity specifically against Firmicute and Actinobacteria. While this might be true, in this work the terminal CPD enzymes from Staphylococcus aureus (Firmicute) were investigated and docking was performed on the ChdC of Listeria monocytogenes (Firmicute). There are well known structural and functional differences in actinobacterial ChdCs (a catalytically important distal histidine is lacking in firmicute ChdCs, https://pubmed.ncbi.nlm.nih.gov/32440366/). Potentially the identified substances interact differently with actinobacterial enzymes.

We have changed the text of the sentence to read:  “Above we have detailed the development of an efficient, robotic, scalable growth assay-based screen to identify compounds that possess antimicrobial activity against monoderm bacteria, in Firmicute species.”

Reviewer 2 Report

The manuscript by Jackson et al describes a newly developed drug screening method exploiting the CPD heme biosynthesis pathway as potential drug target. To this end, the authors created a genetically modified E. coli strain, in which the genes encoding the three terminal enzymes of the PPD pathway were exchanged against those of the CPD pathway. Additionally, the E. coli cgdH was also deleted to avoid the accumulation of protoporphyrin IX. For the drug screening a strain mixture of wt and engineered E. coli was used in a competition assay. With this approach several substances were identified, which specifically inhibited growth of the engineered strain. This growth defect was rescued by heme addition to the medium, in line with the CPD pathway being the target. Finally, in silico docking of some of the substances into the structure of coproheme decarboxylase substantiates the proposal that this enzyme might be the target.

The manuscript is very well written. The data are convincing and presented in a well-structured manner. The approach of using an engineered E. coli strain for initial drug screening instead of the pathogens is very attractive and a very nice model system.

There are only very few minor points that should be addressed:

1.       Abstract, line 13: coproheme decarboxylase

2.       Introduction, line 9: Inhibition of heme biosynthesis pathway enzymes………

3.       Introduction, line 13: add a reference for the small colony variants of S. aureus

4.       Figure 1: the red circle highlighting the vinyl group of protoporphyrinogen IX might be placed a little bit better around the vinyl group

5.       Page 2, second line from bottom: the SHD (ahb) pathway is not present in denitrifiers

6.       Page 3, first line: ……is metabolized by demethylation, deacetylation and then decarboxylation…….

7.       Page 8: In the description of the preparation of the strain mixture for the competition screen, it is stated that a stoichiometric mixture was set up, but different final OD600 values were given (0.1 for SA-CPD and 0.02 for WC). Why is that?

8.       Table 1: the SA-CPD strain produces much more porphyrins and less heme than the parental WC strain indicating that the flow through the whole pathway is impaired in the engineered strain. Is the accumulated porphyrin mainly coproporphyrin III or partially decarboxylated uroporphyrin III? Which enzyme might be the bottleneck and why?

Author Response

Reviewer #2

The manuscript by Jackson et al describes a newly developed drug screening method exploiting the CPD heme biosynthesis pathway as potential drug target. To this end, the authors created a genetically modified E. coli strain, in which the genes encoding the three terminal enzymes of the PPD pathway were exchanged against those of the CPD pathway. Additionally, the E. coli cgdH was also deleted to avoid the accumulation of protoporphyrin IX. For the drug screening a strain mixture of wt and engineered E. coli was used in a competition assay. With this approach several substances were identified, which specifically inhibited growth of the engineered strain. This growth defect was rescued by heme addition to the medium, in line with the CPD pathway being the target. Finally, in silico docking of some of the substances into the structure of coproheme decarboxylase substantiates the proposal that this enzyme might be the target.

The manuscript is very well written. The data are convincing and presented in a well-structured manner. The approach of using an engineered E. coli strain for initial drug screening instead of the pathogens is very attractive and a very nice model system.

There are only very few minor points that should be addressed:

  1. Abstract, line 13: coproheme decarboxylase

Text been changed to coproheme decarboxylase.

  1. Introduction, line 9: Inhibition of heme biosynthesis pathway enzymes………

The word “biosynthesis” has been added.

  1. Introduction, line 13: add a reference for the small colony variants of S. aureus

Reference has been added.

  1. Figure 1: the red circle highlighting the vinyl group of protoporphyrinogen IX might be placed a little bit better around the vinyl group

The figure has been remade and the red circle has been placed better around the vinyl group.

  1. Page 2, second line from bottom: the SHD (ahb) pathway is not present in denitrifiers

The inclusion of “denitrifiers” was removed as suggested.

  1. Page 3, first line: ……is metabolized by demethylation, deacetylation and then decarboxylation…….

Text has been changed as suggested.

  1. Page 8: In the description of the preparation of the strain mixture for the competition screen, it is stated that a stoichiometric mixture was set up, but different final OD600 values were given (0.1 for SA-CPD and 0.02 for WC). Why is that?

The previous sentence provides the explanation.  It reads “SA-CPD bacteria grow more slowly than WC, so the initial inoculum for the screen is created with a higher starting concentration of SA-CPD compared to WC.”   The SA-CPD bacteria grow well, they just grow a little more slowly.  We experimentally determined that starting with a slightly higher amount of SA-CPD compared to WC (ratio/stoichiometry by OD600 absorbance) with our incubation time/conditions works well and reproducibly, and compensates for the difference.  Near the end of a 16-18 hour incubation period in the wells, the bacteria will be close to stationary phase.  There is competition between the bacteria for resources.  Increasing the starting amount of SA-CPD bacteria is enough to allow both strains to grow well in competition, when uninhibited.  The description in the Methods section describes the SOP.

  1. Table 1: the SA-CPD strain produces much more porphyrins and less heme than the parental WC strain indicating that the flow through the whole pathway is impaired in the engineered strain. Is the accumulated porphyrin mainly coproporphyrin III or partially decarboxylated uroporphyrin III? Which enzyme might be the bottleneck and why?

We added the following text: “Assay of porphyrins, protoporphyrin IX, uroporphyrin, heptacarboxylporphyrin, hexacarboxylporphyrin, pentacarboxylporphyrin and coproporphyrin, showed that cgdH KO dramatically reduced PPIX buildup in the replacement mutant strain (Table 1).  Total porphyrins remain elevated in the mutant.  Profiles of individual porphyrins (percent of total) generally resemble that of wild type (approximately 25% coproporphyrin), with a small elevation in partially decarboxylated forms (data not shown).  The peak for coprohemin (iron coproporphyrin) was very small in mutant bacteria.  Because our porphyrin assay oxidizes porphyrinogens to porphyrins prior to UPLC analysis, we do not know whether CgoX or CpfC has become the bottleneck.  It is likely that the reaction catalyzed by CgoX will occur spontaneously, albeit at a slower rate, so we expect that S. aureus CpfC is the bottleneck in our mutant strain of E. coli.  The cgdH knockout noticeably increased the robustness of our screening strain.”

Reviewer 3 Report

This manuscript describes an in vivo screening platform that allows for the rapid and efficient identification of potential anti-microbials that target monoderm, but not diderm, bacteria. Specifically, the screen the authors develop was used to identify compounds that target the heme biosynthesis pathway utilized by monoderm bacteria, which differs in key steps from the pathway identified in diderm bacteria. The screen is elegantly simple, well-described, and ultimately successful. In addition, the authors include internal controls that boost the reliability of the results and highlight the thoughtful nature of the experimental design. Overall, the manuscript as prepared is of high quality and should be of interest to many readers. Some minor suggestions are as follows:

1. The title in a way seems to downplay the results of the manuscript. Would the authors consider editing the title in a way that reflects the successfulness of their approach (rather that just "Screening for anti-microbials")? Also "novel pathway" is a bit distracting to the mission of the manuscript, which is to target an already established pathway.

2. The manuscript, while generally being very descriptive, is a bit weighed down by the plethora of abbreviations, acronyms, protein/gene names, strain names, etc. in addition to the extra text needed to constantly explain all of these things. Could the authors find a way to simplify at least some of these things that that reader needs to remember? For instance, the 2017 Microbiology and Molecular Biology Reviews paper by some of these same authors clearly name these heme synthesis enzymes, yet this paper uses both the old and the new abbreviations (for example, HemQ vs ChdC). In the same vein, on page 11, the SA-CPD is given three different names (SA-CPD, S. aureus hemYHQ mutant with cdgH KO, and SYHQ-NKAN1). Lastly, "SA-CPD" and "WC" are not intuitive for the reader; the reader must remember that SA-CPD is from a strain that produces YFP (and the SA is confusing to the heme audience that thinks of SA as succinyl acetone). Perhaps Sa-CPD-YFP and WT-CFP, or something more descriptive?

3. There were two small sections that could potentially use more references:

a)The first is the paragraph describing the CPD pathway on page 3; at minimum, it is recommended to cite the 2017 paper above which names the pathway, but it would also be advisable to cite some of the literature describing the mechanistic and structural work, particularly with ChdC. By relying solely on reference 20, it seems as though there is not much interest in this pathway when it is in fact of interest to many in the field. 

b) The methods section 2.5 regarding Z' factors does not have a reference; the same reference can be used (maybe?) on page 12 when the authors report that "Z' factors above 0.5 are considered excellent".

4. Figures 6 and 10 are small, and the colors in figure 10 (especially the yellow) are difficult to discern.

5. The compounds identified in the initial screen appeared to specifically target ChdC and not CogX or CpfC. Could the authors comment on whether or not their assay is biased in any way towards identifying ChdC inhibitors?

Author Response

Reviewer #3 comments

Comments and Suggestions for Authors

The experiments are well designed and the results are solid and reliable. The manuscript is well written and readable. I think that this interesting work would certainly advance our understanding of the screening for anti-microbial targeting a novel pathway in heme biosynthesis. I strongly recommend this paper for publication in the Journal. However, I raise some concerns that need to be addressed before publication. If those concerns are adequately address in the revised manuscript, this interesting report would be significantly strengthened.

Concerns that need to be addressed before publication.

The second author is now deceased (*) and the corresponding author designation has been changed to ‡.

“List of Abbreviations” should be incorporated before (or after) the main text. For example, heme, an iron-containing protoporphyrin IX, ALA, 5-aminolevulinate; GtrR, glutamyl tRNA reductase, PPIX, protoporphyrin IX, et al.  What is AlaS at second para, line 9? I think it wrong that heme is ferrous-protoporphyrin IX, but hemin is ferric-protoporphyrin IX. Please sort out names of heme-related compounds. 

A list of abbreviations has been included before the main text:

List of Abbreviations.  PPD, protoporphyrin-dependent pathway, the “classical” heme biosynthesis pathway; CPD, coproporphyrin-dependent pathway, using different biosynthetic intermediates compared to “classical”; heme, an iron-containing protoporphyrin IX; coproheme, an iron containing coproporphyrin, also referred to as iron coproporphyrin or Fe coproporphyrin, an intermediate in the CPD pathway, must undergo decarboxylation to form heme; hemin, the chloride salt of oxidized heme; SA-CPD, E. coli (diderm) mutant with three terminal heme biosynthesis enzymes replaced with CPD forms from S. aureus (monoderm) and expressing YFP fluorophore;  WC, wild type E. coli expressing CFP fluorophore; Enzyme abbreviations are provided and illustrated in Figure 1.

AlaS has been defined as 5-aminolevulinic acid synthase in the text and Figure 1 A.

Page 1, 1. Introduction:   Please cite reviews of heme’s novel functions such as Chem. Rev. 2015, 115, 6491 for heme-based gas sensors, Chem. Soc. Rev. 2019, 48, 5624 for heme-responsive sensors and Chem. Rev. 2018, 118, 6573 for heme-induced oxygen activation at the initial part (first 4 lines) in 1. Introduction. For general introduction of heme, references 1-4 appear not sufficient and to biased to only pathogenic bacteria. 

These references have been added into the introduction.

New table(s): Many genes involved heme biosynthesis are described in this manuscript.  I would suggest the authors to make a table that summarizes the genes’ names and their (presumed) functions as the Supplemental Table. Otherwise, a new Figure that summarizes genes involved in each step of heme biosynthesis and aims of the present research might be beneficial for general readers to grasp the important points in this interesting manuscript. Figure 1 is not sufficient, more detailed explanation would be needed to be attractive for readers.

To address this comment and to make the paper more accessible we have created a new part to Figure 1.  Figure 1A depicts the early part of the heme biosynthesis pathway.  Figure 1B depicts the last three steps immediately relevant to this paper, and provides enzyme, gene and protein names or abbreviations (where applicable), illustrating their presumed functions.

Page 9, 2nd para. Lines 6-10:  I feel curious why the author did not consider the Soret band, because Soret band is much stronger than those in the visible region. The Soret band is used at Page 7, line 7. 

The assay uses the fluorescence of the YFP or CFP stably integrated into the bacterial genome. I think the reviewer is suggesting that we use the optical properties of the substrate(s) coproheme, PPIX, as a readout.  Our screen is based on the actual growth of the bacteria not on the flow of intermediates through the pathway.  The excitation/emission wavelengths provided are for the integrated fluorescent proteins.

Page 10, Table 1, Supplemental Figure 4:  hemin is used. On the other hand, page 15, Figure 8: heme is used. Do the authors recognize the difference between hemin and heme?

Figure 8 legend has been edited to indicate that hemin was added, rather than heme.  Reviewer may notice a color change in these control plots.  Plots had been color matched to Figure 7.  However, one of the four plots was taken from a separate non-matching file and this legend had been used.  Figure and legend colors have been edited and correctly indicate control conditions and readings.

Page 13, Figure 6:   Many drugs/chemicals are used. Figure 6 partially explains the chemical structures of selected drugs/chemicals. Please make a Supplemental Figure describing/sorting out the names and chemical structures of all drugs/chemicals studied in the present work.

We have created a supplemental table with images of drugs/chemicals used and mentioned in the paper.

Page 17, Figure 10: Protein structures, A, B, C should be largely expanded and be more cleared. It is difficult to figure out what the authors want to emphasize in the present figures.

We have removed Figures 10A & 10B and placed them in a new supplemental figure 7.  We have enlarged Figure 10C to form a Figure 10 to contain a single cleared image.  We have edited the Figure legend to point out that we wish to emphasize that there is reasonable space within the binding pocket to accommodate prodiginine molecules. 

Page 17, 5. Conclusions: It is difficult to grasp the whole story and significant results obtained in this interesting study.  I would suggest the authors to expand this section by incorporating a new cartoon figure with chemical structures of inhibitors that describes important points obtained in the study. This cartoon is crucial because general readers tend to read only abstract, figures and conclusions without reading details of the paper.

We have created a summary cartoon: Figure 11 that describes important points from the study.

It appears that several groups made one or more sections separately but failed to integrate the writing system. A specific group or person should be responsible to unify the manuscript throughout for general readers to easily grasp the points at a glance.

In summary, I strongly recommend publication of this great paper after minor concerns are adequately addressed. 

Reviewer 4 Report

The experiments are well designed and the results are solid and reliable. The manuscript is well written and readable. I think that this interesting work would certainly advance our understanding of the screening for anti-microbial targeting a novel pathway in heme biosynthesis. I strongly recommend this paper for publication in the Journal. However, I raise some concerns that need to be addressed before publication. If those concerns are adequately address in the revised manuscript, this interesting report would be significantly strengthened.

Concerns that need to be addressed before publication.

** Correspondence, but NOT * Correspondence.  The last author did not decease.

“List of Abbreviations” should be incorporated before (or after) the main text. For example, heme, an iron-containing protoporphyrin IX, ALA, 5-aminolevulinate; GtrR, glutamyl tRNA reductase, PPIX, protoporphyrin IX, et al.  What is AlaS at second para, line 9? I think it wrong that heme is ferrous-protoporphyrin IX, but hemin is ferric-protoporphyrin IX. Please sort out names of heme-related compounds. 

Page 1, 1. Introduction:   Please cite reviews of heme’s novel functions such as Chem. Rev. 2015, 115, 6491 for heme-based gas sensors, Chem. Soc. Rev. 2019, 48, 5624 for heme-responsive sensors and Chem. Rev. 2018, 118, 6573 for heme-induced oxygen activation at the initial part (first 4 lines) in 1. Introduction. For general introduction of heme, references 1-4 appear not sufficient and to biased to only pathogenic bacteria. 

New table(s): Many genes involved heme biosynthesis are described in this manuscript.  I would suggest the authors to make a table that summarizes the genes’ names and their (presumed) functions as the Supplemental Table. Otherwise a new Figure that summarizes genes involved in each step of heme biosynthesis and aims of the present research might be beneficial for general readers to grasp the important points in this interesting manuscript. Figure 1 is not sufficient, more detailed explanation would be needed to be attractive for readers.

Page 9, 2nd para. Lines 6-10:  I feel curious why the author did not consider the Soret band, because Soret band is much stronger than those in the visible region. The Soret band is used at Page 7, line 7. 

Page 10, Table 1, Supplemental Figure 4:  hemin is used. On the other hand, page 15, Figure 8: heme is used. Do the authors recognize the difference between hemin and heme?

Page 13, Figure 6:   Many drugs/chemicals are used. Figure 6 partially explains the chemical structures of selected drugs/chemicals. Please make a Supplemental Figure describing/sorting out the names and chemical structures of all drugs/chemicals studied in the present work.

Page 17, Figure 10: Protein structures, A, B, C should be largely expanded and be more cleared. It is difficult to figure out what the authors want to emphasize in the present figures.

Page 17, 5. Conclusions: It is difficult to grasp the whole story and significant results obtained in this interesting study.  I would suggest the authors to expand this section by incorporating a new cartoon figure with chemical structures of inhibitors that describes important points obtained in the study. This cartoon is crucial because general readers tend to read only abstract, figures and conclusions without reading details of the paper.

It appears that several groups made one or more sections separately but failed to integrate the writing system. A specific group or person should be responsible to unify the manuscript throughout for general readers to easily grasp the points at a glance.

In summary, I strongly recommend publication of this great paper after minor concerns are adequately addressed. 

Author Response

Reviewer #4

This manuscript describes an in vivo screening platform that allows for the rapid and efficient identification of potential anti-microbials that target monoderm, but not diderm, bacteria. Specifically, the screen the authors develop was used to identify compounds that target the heme biosynthesis pathway utilized by monoderm bacteria, which differs in key steps from the pathway identified in diderm bacteria. The screen is elegantly simple, well-described, and ultimately successful. In addition, the authors include internal controls that boost the reliability of the results and highlight the thoughtful nature of the experimental design. Overall, the manuscript as prepared is of high quality and should be of interest to many readers. Some minor suggestions are as follows:

  1. The title in a way seems to downplay the results of the manuscript. Would the authors consider editing the title in a way that reflects the successfulness of their approach (rather that just "Screening for anti-microbials")? Also "novel pathway" is a bit distracting to the mission of the manuscript, which is to target an already established pathway.

We have changed the title to:  Exploiting differences in heme biosynthesis between bacterial species to screen for novel antimicrobials

  1. The manuscript, while generally being very descriptive, is a bit weighed down by the plethora of abbreviations, acronyms, protein/gene names, strain names, etc. in addition to the extra text needed to constantly explain all of these things. Could the authors find a way to simplify at least some of these things that that reader needs to remember? For instance, the 2017 Microbiology and Molecular Biology Reviews paper by some of these same authors clearly name these heme synthesis enzymes, yet this paper uses both the old and the new abbreviations (for example, HemQ vs ChdC). In the same vein, on page 11, the SA-CPD is given three different names (SA-CPD, S. aureus hemYHQ mutant with cdgH KO, and SYHQ-NKAN1). Lastly, "SA-CPD" and "WC" are not intuitive for the reader; the reader must remember that SA-CPD is from a strain that produces YFP (and the SA is confusing to the heme audience that thinks of SA as succinyl acetone). Perhaps Sa-CPD-YFP and WT-CFP, or something more descriptive?

We have removed the older abbreviations for genes and only use the new abbreviations.  We have changed SA-CPD and WC throughout the paper to Sa-CPD-YFP and WT-CFP as suggested.  As we made these changes we changed “Total Porphyrins” in Figure 9 and Supplemental Figure 4 to Σ87654 porphyrins (intermediate porphyrins with 8, 7, 6, 5 & 4 carboxyl groups) to be more clear about which porphyrins we were measuring. 

  1. There were two small sections that could potentially use more references:

a)The first is the paragraph describing the CPD pathway on page 3; at minimum, it is recommended to cite the 2017 paper above which names the pathway, but it would also be advisable to cite some of the literature describing the mechanistic and structural work, particularly with ChdC. By relying solely on reference 20, it seems as though there is not much interest in this pathway when it is in fact of interest to many in the field. 

Additional references have been added to address this and other reviewers’ comments.

  1. b) The methods section 2.5 regarding Z' factors does not have a reference; the same reference can be used (maybe?) on page 12 when the authors report that "Z' factors above 0.5 are considered excellent".

The following reference has been added to address this concern,  Zhang, JH; Chung, TDY; Oldenburg, KR (1999). "A simple statistical parameter for use in evaluation and validation of high throughput screening assays". Journal of Biomolecular Screening4 (2): 67–73. doi:10.1177/108705719900400206. PMID 10838414. S2CID 36577200

  1. Figures 6 and 10 are small, and the colors in figure 10 (especially the yellow) are difficult to discern.

Figure 10 has been separated into a main figure and supplemental figure, allowing for the enlargement of the image.  A supplemental table 3 has been created to individually include more chemical structures, and enlarge the structures (found in figure 6) for better viewing.

  1. The compounds identified in the initial screen appeared to specifically target ChdC and not CgoX or CpfC. Could the authors comment on whether or not their assay is biased in any way towards identifying ChdC inhibitors?

We do not have reason to believe that the assay is biased towards identifying ChdC inhibitors.  Based on UPLC analysis, our two strongest hits do appear to target ChdC.  However, we mention miconazole nitrate (a weaker hit) simply because of its known antibacterial activity and because of the known activity of azole drugs against Mycobacterium.

Additional Comments:

Finally, the E coli proto oxidase is named PgdH1, rather than PgoX.  We have changed our manuscript to reflect that.